# Real-Time Visualization of Cytosolic and Mitochondrial ATP Dynamics in Response to Metabolic Stress in Cultured Cells

**DOI:** 10.3390/cells12050695

**Published:** 2023-02-22

**Authors:** Donnell White, Lothar Lauterboeck, Parnia Mobasheran, Tetsuya Kitaguchi, Antoine H. Chaanine, Qinglin Yang

**Affiliations:** 1Cardiovascular Center of Excellence, Louisiana State University Health Sciences Center, New Orleans, LA 70112, USA; 2Department of Pharmacology and Experimental Therapeutics, School of Graduate Studies, Louisiana State University Health Sciences Center, New Orleans, LA 70112, USA; 3School of Medicine, Louisiana State University Health Sciences Center, New Orleans, LA 70112, USA; 4Cell Biology, Life Science Solutions, Thermo Fisher Scientific, Frederick, MD 21702, USA; 5Laboratory for Chemistry and Life Science, Institute of Innovative Research, Tokyo Institute of Technology, 4259 Nagatsuta-cho, Midori-ku, Yokohama 226-8503, Kanagawa, Japan; 6HealthPartners Group, Regions Hospital, Saint Paul, MN 55101, USA

**Keywords:** mitochondria, ATP, biosensor, fluorescence, metabolism

## Abstract

Adenosine 5′ triphosphate (ATP) is the energy currency of life, which is produced in mitochondria (~90%) and cytosol (less than 10%). Real-time effects of metabolic changes on cellular ATP dynamics remain indeterminate. Here we report the design and validation of a genetically encoded fluorescent ATP indicator that allows for real-time, simultaneous visualization of cytosolic and mitochondrial ATP in cultured cells. This dual-ATP indicator, called smacATPi (simultaneous mitochondrial and cytosolic ATP indicator), combines previously described individual cytosolic and mitochondrial ATP indicators. The use of smacATPi can help answer biological questions regarding ATP contents and dynamics in living cells. As expected, 2-deoxyglucose (2-DG, a glycolytic inhibitor) led to substantially decreased cytosolic ATP, and oligomycin (a complex V inhibitor) markedly decreased mitochondrial ATP in cultured HEK293T cells transfected with smacATPi. With the use of smacATPi, we can also observe that 2-DG treatment modestly attenuates mitochondrial ATP and oligomycin reduces cytosolic ATP, indicating the subsequent changes of compartmental ATP. To evaluate the role of ATP/ADP carrier (AAC) in ATP trafficking, we treated HEK293T cells with an AAC inhibitor, Atractyloside (ATR). ATR treatment attenuated cytosolic and mitochondrial ATP in normoxia, suggesting AAC inhibition reduces ADP import from the cytosol to mitochondria and ATP export from mitochondria to cytosol. In HEK293T cells subjected to hypoxia, ATR treatment increased mitochondrial ATP along with decreased cytosolic ATP, implicating that ACC inhibition during hypoxia sustains mitochondrial ATP but may not inhibit the reversed ATP import from the cytosol. Furthermore, both mitochondrial and cytosolic signals decrease when ATR is given in conjunction with 2-DG in hypoxia. Thus, real-time visualization of spatiotemporal ATP dynamics using smacATPi provides novel insights into how cytosolic and mitochondrial ATP signals respond to metabolic changes, providing a better understanding of cellular metabolism in health and disease.

## 1. Introduction

ATP is the “energy currency” of all living organisms. Consistent cellular ATP is vital in maintaining and regulating normal mammalian cells. Under typical conditions, mitochondrial oxidative phosphorylation (OXPHOS) produces over 90% of ATP, while cytosolic glycolysis yields the remaining portion. Cellular ATP deficit has been well recognized as a cause of numerous diseases [1]. When cells are proliferating, stressed, or have a limited oxygen supply, glycolysis emerges as the primary source of ATP. This metabolic flexibility serves the purpose of maintaining stable cellular ATP. Under hypoxic conditions, mitochondria consume ATP, presumably from cytosolic glycolysis, to keep the mitochondrial membrane potential through the reverse pumping of ATP synthase with ATP hydrolysis. However, the cytosolic and mitochondrial ATP transportation and dynamics in response to metabolic changes and hypoxia remain obscure.

With ATP production and consumption being an essential aspect of understanding the basics of metabolic pathologies, imaging of real-time cytosolic and mitochondrial ATP changes within single cells provides crucial insights into regulatory mechanisms by which cells survive metabolic disturbances. Recent development has shown improvement in ATP imaging technology by utilizing various starting materials, such as small organic indicators, nanoparticles, and fluorescent probes [2]. Numerous methodologies have been developed to visualize ATP in cells using fluorescence resonance energy-transfer-based genetically encoded indicators. These assess ATP dynamics in mitochondria, the cytosol, and the endoplasmic reticulum [3,4]. Measuring total ATP levels within cellular compartmental pools in real time is a newer and more innovative approach to qualitatively analyzing ATP. This method is semi-quantitative and can help determine changes in ATP concentrations in one region of a cell compared to another in various disease states. The currently developed technologies that utilize this approach to ATP quantification are mainly genetically encoded biosensors [5,6,7]. Many of these genetically encoded biosensors are based on the ATP-dependent conformational change of the *B. subtilis* ϵ subunit [1,2]. The *B. subtilis* ϵ subunit is linked with various forms of fluorescent proteins. ATP binding alters the conformation and the environment surrounding the fluorescent protein chromophore [3], leading to illumination, indicating the presence of ATP. While this strategy has a proven record of success in indicating cellular ATP, it remains unclear if the existence of a *B. subtilis* ϵ subunit in mammalian cells would interfere with cellular respiration, which may lead to misinterpretation. Furthermore, dual or triple ATP sensors that simultaneously detect ATP in different cellular compartments with a single transfection are desirable to gain in-depth insights into energetic cellular activities.

This study demonstrates the feasibility of a novel genetically encoded ATP biosensor that renders the spatiotemporal evaluation of ATP changes in mitochondria and the cytosol. By utilizing this dual ATP biosensor, smacATPi (simultaneous mitochondrial and cytosolic ATP indicator), we explore previously unanswered questions regarding the spatiotemporal ATP transportation and dynamics in response to various metabolic stresses. 

## 2. Materials and Methods

### 2.1. Design and Validation of an In Vitro Dual-ATP Indicator (smacATPi)

Individual ATP indicators that allow for the evaluation of cytosolic and mitochondrial ATP were created by inserting the ATP-binding region of the epsilon subunit of bacterial F_1_F_o_-ATP synthase into the GFP citrine variant and the red fluorescent mApple protein [8]. These single indicators were created by the Kitaguchi lab and were entitled Monitoring ATP Level intensity-based turn-on indications, or MaLions [8]. When ATP binds to the epsilon subunit, the ATP-dependent conformation of the subunit allows for the tertiary formation of the fluorophore to occur, indicating that ATP is present. To evaluate mitochondrial ATP, a mitochondrial signaling sequence (MSS) of the subunit VIII of human cytochrome c oxidase (COXVIII) was added to the red MaLion (MaLionR), and the green MaLion indicates ATP levels in the cytosol (MaLionG).

In this study, we designed and validated a dual-fluorophore ATP indicator that monitors cytosolic and mitochondrial ATP in real-time. Our lab combined these two individual indicators previously discussed into one vector, calling this dual indicator system smacATPi (simultaneous mitochondrial and cytosolic ATP indicator). The plasmid we designed contains the two single indicator sequences (hereon called mito-smacATPi and cyto-smacATPi) separated by P2A, a self-cleaving peptide, ensuring proper translation and separation of the fluorophores.

### 2.2. Generation and Validation of neg-smacATPi

Two point mutations were induced in the plasmid at the regions where the proteins are linked. This was done to validate that the fluorescence seen by smacATPi is not due to autofluorescence. Two small mutations were induced into the smacATPi plasmid using the Q5^®^ Site-Directed Mutagenesis Kit (New England BioLabs, Ipswich, MA, USA). The recommended website, http://nebasechanger.neb.com/ (accessed on 3 May 2021), was used to design the best primers for SDM (site-directed mutation) deletions to create neg-smacATPi. A 21-nucleotide sequence (5′ CAAGGAGGACGGCAACATCCT 3′) was designed to be removed from the plasmid region that codes for the citrine protein.

Additionally, a 15-nucleotide sequence (5′ CGGCGCCCTGAAGAG 3′) was designed to be removed from the mApple region of the plasmid. After inducing these deletions within the plasmid, the fluorescent proteins were expected not to maintain their structure and conformation to fluoresce. The primers that induced SDM in the cyto-smacATPi linker region of the plasmid were F: GGGGCACAAGCTGGAGTA and R: AAGTCGATGCCCTTCAGC. The primers used to induce SDM in the mito-smacATPi linker region of the plasmid were F: CGAGATCAAGAAGGGGCTGAG and R: TCCTCGGGGTACATCCGC. The mutations were confirmed via Sanger Sequencing and transfection (Appendix A).

### 2.3. Cell Culture

HEK293T cells were obtained from ATCC. The cells were maintained at 37 °C, 5% CO_2_ environment, and cultured in Dulbecco’s Modified Eagle Medium + Glutamax (Gibco, Billings, MA, USA), supplemented with filtered 10% Fetal Bovine Serum (Atlanta Biologics, Flowery Branch, GA, USA) and 1% penicillin/streptomycin (Gibco, MA, USA). The cells were detached using 0.25% Trypsin-EDTA (1X) (Gibco, MA, USA), and plated in 96-well black/clear bottom plates (Thermo Fisher, Waltham, MA, USA) at a density where the cells reach ~70% for transfection 24 h later.

### 2.4. HEK293T Cell Transfection

Transfection was optimized in HEK293T cells using Fugene 6 transfection reagent (Promega, Madison, WI, USA). HEK293T cells were cultured to a density of ~70% on the day of transfection. Our smacATPi and neg-smacATPi plasmids were transfected into HEK293T cells using Fugene6 transfection reagent (Promega, WI, USA) following the manufacturer’s instructions. At 24 h post-transfection, cells were imaged during drug trial runs and analysis. The transfection efficiency of the smacATPi indicator for HEK293T cells was ~80–90%.

### 2.5. Mouse Embryonic Fibroblast (MEF) Isolation

Embryos were harvested from female C57/BJ mice 12.5–14 days after the appearance of the copulation plug. The pregnant female was euthanized according to protocol, then under a sterile hood, cut with scissors to expose the abdominal wall. The uterine horns were cut away from the uterus. The uterus was placed into a petri dish with PBS, and the embryos were removed by slicing through the uterus in the region between each embryo. The embryos were transferred into a new fresh PBS petri dish to remove blood. The head was cut off above the eyes to remove neural tissues. The red tissues were removed, and the embryo was washed again in PBS and placed into Trypsin/EDTA. They were placed into a cell culture incubator for 10 min. The cells were then moved to a conical tube, and MEF medium was added. The cells were allowed to settle, and then the supernatant consisting of single cells and cell clusters was transferred to a 10 cm culture dish. The following morning, the media was replaced. Cells were used for electroporation once they reached confluence. All experimental mouse procedures were approved by the Institutional Animal Care and Use Committee of Louisiana State University Health Science Center-New Orleans.

### 2.6. Electroporation of MEF Cells

Around 200,000 MEF cells were suspended in 300 μL of OptiMem (Gibco, MA, USA) and placed into a 4 mm Gene Pulser cuvette (Bio-Rad, Hercules, CA, USA). The electroporation conditions were: 250 V, two pulses, each 10 ms in length, and 10 s intervals in a 4 mm cuvette. This was done at room temperature with 10 µg smacATPi DNA. After the electroporation, cells were plated in a 96-well plate with growth media for maximum cellular survival. The cells were checked 24 h post-electroporation to determine transfection efficiency.

### 2.7. Isolation and Culture of Adult Rat Cardiomyocytes (ARCM)

Male and female Sprague-Dawley rats weighing 250–300 g are given sodium heparin (200 U, Sigma Aldrich, St. Louis, MO, USA) intraperitoneally. Twenty min later, the animal was anesthetized with 30% isoflurane via a nose cone until breathing cessation. Once the animal was anesthetized (no evidence of reflex with toe pinch), the cervical spine was dislocated, the chest was opened, and the heart was immediately removed and placed into ice-cold PBS (Gibco, MA, USA). The heart was attached quickly to a cannula of a Langendorff apparatus. The heart was perfused with perfusion buffer (Krebs–Henseleit Buffer, KHB) for 4 min and then perfused with a digestion buffer consisting of KHB, collagenase type II, and trypsin. After digested for 15–20 min, the heart was removed from the cannula, the atria were removed, and scissors and forceps were used to cut and shake the ventricles in the digestion solution. After 1–2 min, a stop solution was added to inhibit further digestion. The cardiomyocytes were filtered into a 50 mL tube, and then CaCl_2_ (Sigma Aldrich, MO, USA) was reintroduced for 20 min. After calcium reintroduction, the cells were re-suspended in growth media and plated for lentivirus transduction on 96-well black/clear bottom plates (Thermo Fisher, MA, USA) plates coated with 40 µg/mL laminin (Invitrogen, Thermo Fisher, USA). All rat procedures were approved by the Institutional Animal Care and Use Committee of Louisiana State University Health Science Center–New Orleans.

### 2.8. Generation of Adenovirus for smacATPi

An adenovirus was made with smacATPi by VectorBuilder (Chicago, IL, USA) to apply this technique in hard-to-transfect cell lines other than HEK293T. To start, 20,000 adult rat cardiomyocytes were seeded onto a 6-well plate coated with 40 µg/mL of laminin (Invitrogen, Thermo Fischer, USA) and incubated at 37 °C and 5% CO_2_ for one hour to allow adherence of the cells. The adenovirus (200 multiplicity of infection) was added directly to the cell media. The fluorescence intensity of the cells was evaluated every 24 h, and the maximum transduction efficiency was observed 72 h post-infection.

### 2.9. Drug Selection

To investigate cytosolic and mitochondrial ATP dynamics in response to cellular metabolic activation and inhibition, we treated cultured HEK293T cells with metabolic effectors. Various concentrations of the drugs used were tested (Appendix A). 2-Deoxy-d-glucose (2-DG, Sigma Aldrich, MO, USA) is a glycolysis inhibitor, and it was given at a concentration of 25 mm. Oligomycin (Sigma Aldrich, MO, USA), an electron transport chain (ETC) complex V inhibitor, was given at a concentration of 100 µM and decreased the ability of live cells to produce adequate ATP needed for cellular metabolism. To determine how inhibiting the ADP/ATP exchange between cytosol and mitochondria will affect ATP dynamics and trafficking within the cell, we treated cultured HEK293T cells with Atractyloside (ATR, Cayman Chemical, Ann Arbor, MI, USA), an inhibitor of the ATP/ADP carrier within the inner mitochondrial membrane. ATR was given at a concentration of 100 µM, and cell imaging was continuous before and after treatment. These same concentrations were used in hypoxia experiments. Oligomycin stock solution was prepared in ethanol and then diluted in PBS for working solution. 2-DG and ATR were dissolved in water for their respective stock solutions, then diluted in PBS for their final drug concentrations.

### 2.10. Evaluation of Mitochondrial Membrane Potential

Tetramethylrhodamine (TMRM), a mitochondrial membrane potential indicator, was used alone and in conjunction with certain drugs to evaluate their effects on the membrane potential. The cells were plated at a density of 30,000 cells per well in a 96-well plate and allowed to grow overnight until ~90% confluency. A concentration of 50 nm TMRM (Image-IT TMRM Reagent, Thermo Fischer Scientific, Waltham, MA, USA) was added to each well, and the plate was incubated in the dark for 30 min. The Cytation5 Cell Imaging Multi-Mode Reader (BioTek, Winooski, VA, USA) was used to monitor the mitochondrial membrane potential in both hypoxic and normoxic environments to validate that the conditions within Cytation5 are hypoxic (Appendix A). In order to confirm that TMRM was working correctly, FCCP (15 µM) was used to verify that mitochondrial membrane depolarization shows a decrease in TMRM fluorescence. Our chosen drugs (ATR and 2-DG) were used to investigate the mitochondrial membrane potential after drug administration. TMRM has an absorbance peak of 584 nm, and its emission peak is 574 nm. Imaging conditions were the same as culture conditions, as they were maintained at normoxia at 37 °C in 5% CO_2_, or hypoxia at 37 °C, 2% O_2_, and 5% CO_2_.

### 2.11. Evaluation of Mitochondrial pH Change

A mito-pH indicator was used to investigate mitochondrial pH fluctuations that may interact with our biosensor sensitivity. The pH indicator used was GW1-Mito-pHRed, a gift from Gary Yellen via Addgene (Addgene plasmid #31474, accessed 6 June 2022). A control experiment in normoxia was run post-transfection of HEK293T cells with mito-pH tracker for 2 h, and then an additional run was run in hypoxic conditions for the same time (Appendix A). 

### 2.12. Image Acquisition

The transfected cells were imaged using Cytation5 Cell Imaging Multi-Mode Reader (BioTek, VA, USA). Images and cell runs were recorded at 20× magnification. Imaging conditions were the same as culture conditions, maintained at 37 °C with 5% CO_2_ unless hypoxic conditions were implemented. For green fluorescence, the wavelength used was 488 nm, red was 650 nm, and blue was 461 nm. Supplemental movies are provided, allowing for visualization of real-time change over time. Surface plots were obtained by converting raw fluorescent images into 8-bit, applying FIRE LUT colorization, and then plotting these converted images in ImageJ (National Institutes of Health, Bethesda, MD, USA). 

### 2.13. Hypoxic Conditions for HEK293T Cells

For hypoxia experiments, HEK293T cells were allowed to incubate and were imaged using Cytation5 Cell Imaging Multi-Mode Reader (BioTek, VA, USA) with an environment of 5% CO_2_, 93% N_2_, and 2% O_2_.

### 2.14. Statistical Analysis

The collected image stacks were obtained from Cytation5 Cell Imaging Multi-Mode Reader (BioTek, VA, USA), and imaging analysis was done via software within Cytation5 to analyze and extract fluorescent data. The total intensities were taken from 20× imaging fields of each 96-well, as the variability between cells is too great to measure individually. Each cell field contains a minimum of 10 cells for analysis, and each drug trial/control was done 3–9 times. Therefore, 30–90 cells were analyzed for each experiment. Raw fluorescence intensity values for each run are shown in Appendix A. Change in fluorescence was determined as a ratio of fluorescence intensity over time (F/F_0_). Each drug trial was normalized to its respective PBS vehicle control. Hypoxia experiments were normalized to the 4-h normoxia control in conjunction with mito-pH to account for both fluorescent variability and pH changes (Figure 4D and Appendix A). The normalized F/F_0_ values were plotted with GraphPad Prism 9 statistical software (La Jolla, CA, USA). The endpoint/initial fluorescent intensities were also determined and indicated in bar graphs. All data points on graphs are expressed as mean ± SEM. An unpaired *t*-test was done to determine a significant difference between the end fluorescent value and the initial value in the fluorescent bar graphs, and 2-way ANOVA was done for experiments with three or more variables. The highlighted portions of the graphs indicate the standard errors of the mean. *p*-value indicators are as follows: ns = *p* > 0.05, * = *p* ≤ 0.05, ** = *p* ≤ 0.01, *** = *p* ≤ 0.001, and **** = *p* ≤ 0.0001.

## 3. Results

### 3.1. Design, Generation, and Validation of smacATPi (Simultaneous Mitochondrial and Cytosolic ATP Indicator)

To spatiotemporally visualize cytosolic and mitochondrial ATP dynamics simultaneously within a single cell with a single transfection, we generated a fluorescence-based dual-ATP indicator called simultaneous mitochondrial and cytosolic ATP indicator, hereon called smacATPi. We designed this biosensor by combining two previously described cytosolic and mitochondrial ATP indicators, previously named MaLionG and mitoMaLionR [2], which, respectively, fused the ATP-binding region of the ε subunit of F_1_F_o_-ATP synthase (*B. subtilis*) with green fluorescent protein (GFP) Citrine variant and mApple. A mitochondrial targeting sequence (MTS) from COX II was included in the mitoMaLionR for mitochondrial targeting (Figure 1A). These two fusion proteins were cloned into a single plasmid separated by a short P2A self-cleaving peptide, ensuring the appropriate separation of the proteins (Figure 1A). Once bound to ATP, the protein undergoes a conformational change and emits fluorescence (Figure 1E) [2]. When the above plasmid is transfected in cultured cells, the ATP indicators become localized to the cytosol and mitochondria with different colors (green and red, respectively) (Figure 1B–F). Furthermore, the co-expression of both proteins is high, as out of all transfected cells, 85% are co-transfected, 88% express cyto-smacATPi, and 96% express mito-smacATPi (Appendix A). 

The smacATPi were transfected in cultured HEK293T, adult rat cardiomyocytes, and mouse embryonic fibroblasts. Fluorescent imaging validated that dynamic cytosolic and mitochondrial ATP signals were green and red colors within the respective cellular compartments in both cell types (Figure 1G,H). These results validate the feasibility and utility of smacATPi in proliferative and post-differentiated cells. 

HEK293T cells and adult rat cardiomyocytes with smacATPi expression are broadly comparable to those without smacATPi expression in their growth, morphology, and cell survival. To ascertain that smacATPi expression does not interfere with basal cellular metabolism, we conducted a Cell Mito Stress Test on cultured HEK293T cells using an Extracellular Flux Analyzer (Seahorse, XFe24, Agilent, Santa Clara, CA, USA) to assess critical mitochondrial function (Figure 2). Cultured HEK293T cells with smacATPi expression did not perturb mitochondrial function. Basal respiration (oxygen consumption rate), ATP production, maximal respiration, spare capacity, coupling efficiency, non-mitochondrial oxygen consumption, and protein leak were unchanged (Figure 2). Furthermore, as previously described, a plasmid containing a mutated form of the fluorescent proteins (Appendix A) was used to exclude the possibility that the red and green fluorescent signals are derived from autofluorescence. We confirmed that the mutated form of the indicator showed no fluorescence (Appendix A).

### 3.2. Cytosolic and Mitochondrial ATP Dynamics and Trafficking in Response to Glycolytic and Mitochondrial Inhibitions under Normoxic Conditions

To define how mitochondrial and cytosolic ATP production processes adapt within cells for optimal function and survival, we treated cultured HEK293T cells with drugs that inhibit cytosolic glycolysis and mitochondrial ATP synthesis by investigating how ATP dynamics change in real time. Real-time fluorescent changes were recorded to monitor the flux of dynamic ATP signals in cytosol and mitochondria. The changes in fluorescent intensity of cyto-smacATPi and mito-smacATPi in cultured HEK293T cells treated with the inhibitors were normalized to a PBS vehicle control (Figure 3N and Appendix A). The drugs used were 2-deoxyglucose (2-DG), a glycolysis inhibitor, and oligomycin, a complex V inhibitor (Figure 3A). When treated with 2-deoxyglucose (2-DG), cytosolic ATP signals were significantly decreased within 15 min and plateaued at 30 min post-injection (Figure 3B–K and Appendix A). After normalization to the PBS vehicle control, cytosolic ATP signal decline post-2-DG treatment was substantial, dropping by 30% (Figure 3C). Additionally, there was also a decrease seen in mitochondrial ATP post-2-DG administration (Figure 3C). When treated with oligomycin (100 µM), an ATP synthase (complex V) inhibitor, mitochondrial ATP declined by around 10%. After normalization, the cytosolic ATP signal increased post-oligomycin by 10% (Figure 3L,M and Appendix A). The increase in cyto-smacATPi indicates a compensatory increase in glycolysis to compensate for mitochondrial ATP levels being suboptimal post-oligo administration. These results demonstrate how smacATPi monitors real-time ATP change in two crucial cellular compartments of ATP production and consumption. 

### 3.3. ATP Regulation and Cellular Response in Hypoxia

Utilizing this novel technique, we further investigated a long-puzzling question in the field to investigate cellular ATP distribution in cells under hypoxic conditions and whether reversed ATP transportation is occurring via ADP/ATP Carrier (AAC). AAC is located in the inner mitochondrial membrane and allows for the exchange of free ATP and ADP across the inner mitochondrial membrane (Figure 4A) [3]. Cytosolic-free ADP is transported via the intermembrane space to the mitochondrial matrix. In contrast, ATP produced from OXPHOS is transported from the mitochondrial matrix to the intermembrane space, then to the cytoplasm via voltage-dependent anion channel 1 (VDAC1) [4] (Figure 4A). Under hypoxic conditions, cytosolic glycolysis is upregulated to generate ATP, and cytosolic ATP is thought to reverse-transfer to mitochondrial matrix, while further being hydrolyzed by F_1_F_o_-ATP synthase to pump proton back to the mitochondrial intermembrane to maintain membrane potential and prevent cell death [5] (Figure 4F). We first tested the effects of AAC inhibition in cultured HEK293T cells under normoxia. Atractyloside (ATR) is a glycoside inhibiting ADP and ATP exchange (Figure 4A) [6]. After normalization, it was seen that ATR treatment in normoxic conditions attenuated both cytosolic and mitochondrial ATP signals in cultured HEK293T cells (Figure 4B,C and Appendix A). These results indicate that although there is inhibition of ADP/ATP exchange between the cytosol and mitochondrial compartments, the mitochondrial production does not waiver.

Next, we investigate how cellular ATP changes under hypoxic conditions to determine how energy balance is maintained in the mitochondria and cytosol compartments under hypoxia conditions. HEK293T cells were imaged in hypoxia using a Cytation5 Cell Imaging Multi-Mode Reader (BioTek, VA, US) with 5% CO_2_ and 2% O_2_. Each hypoxic treatment was conducted for 4 h, and a normoxic control run was used to normalize hypoxia experiments (Figure 4D and Appendix A). The MaLion indicators were selected, showing minimal fluorescent damping due to the hypoxia-related intracellular acidity [2]. To further evaluate the effects of pH due to hypoxia on fluorescent variability, a mito-pHRed indicator was used to determine if pH played a role in the fluorescent sensitivity (Appendix A). After normalization to the normoxic mito-pH, pH showed negligible effects on smacATPi fluorescence (Appendix A).

When smacATPi-expressing cells were subjected to hypoxia, mitochondrial ATP signal was profoundly increased (40%) with cytosolic ATP downregulation (25%) (Figure 4G,H and Appendix A). Furthermore, mitochondrial potential decreased in hypoxia conditions, ensuring that the cells were experiencing hypoxic effects (Appendix A).

For hypoxia experiments with drug administration, the cells were allowed to acclimate to the hypoxic environment for 2 h before drug administration, then were monitored for another 2 h (Figure 4E). Atractyloside (ATR) (100 µM) was administered to cells 2 h post-hypoxia exposure (Figure 4I and Appendix A). The ATR-treated cells in hypoxia showed no statistical difference in mitochondrial ATP levels compared to hypoxia alone (Figure 4I,K). However, after ATR treatment, the decline of cytosolic ATP was reversed and remained at similar levels prior to the initiation of hypoxia, suggesting that ATP transport is not necessarily mediated by AAC in similar situations (Figure 4G,I).

The effects of combination therapy, ATR and 2-DG, were also evaluated in hypoxia. In the same way as the previous experiment, ATR (100 µM) and 2-DG (25 mm) were administered to the cells 2 h after initiation of hypoxia exposure (Figure 4J). Four hours after hypoxia initiation, mitochondrial ATP levels dropped, leveling out around initial baseline levels (Figure 4J,K). The addition of 2-DG caused a decrease in cytosolic ATP compared to ATR only, yet slightly higher than hypoxia alone. However, this was not significant compared to either (Figure 4J,K).

## 4. Discussion

ATP is a molecule involved in energy storage and transport within cells. Direct monitoring of cellular ATP distribution dynamics is highly desired, to help better understand the biochemical mechanisms underpinning cellular function in both standard and diseased states. In this report, we demonstrate the feasibility and utilization of a novel fluorescence-based genetic indicator of ATP, simultaneously monitoring real-time ATP changes in the cytosol and mitochondrial compartments of cells.

### 4.1. Feasibility of the Dual ATP Indicator for Real-Time ATP Visualization within the Cytosol and Mitochondria of Cultured Cells

Our current understanding of ATP status in different cellular compartments is mainly based on indirect measurement of ATP on homogenates of cells and tissues using luciferase-based chemiluminescent assays, liquid chromatography, imaging phosphorous-31 (31P) (MRS/MRI), mass spectrometry, or magnesium green [7]. These assays are effective in quantitatively measuring total ATP in samples, but not for visualization of compartmental ATP changes in real time within an individual living cell. Luciferase-based ATP imaging has been developed and is feasible for visualizing cellular ATP [1]. However, this technique is unsuitable for hypoxia-related investigations because the Luciferase-based reaction requires O_2_. Genetically encoded biosensors are a relatively inexpensive approach to providing direct visualization of ATP changes [8]. Numerous fluorescent ATP indicators have been developed [9]. However, these techniques often suffer from high signal-to-noise ratios or require specialized equipment to determine the spatiotemporal dynamics of ATP within cells (see review [10]). MaLions are intensiometric biosensors that fluoresce brighter as more ATP is bound and have been designed to target specific cellular organelles, thus making ATP levels visible across multiple compartments [2]. In the current study, we test the feasibility of a dual ATP indicator to visualize ATP in the cytosolic and mitochondrial compartments of cultured cells. Our results further support the feasibility of revealing mitochondrial and cytosolic ATP abundances and dynamic changes using fluorescent biosensors. 

Moreover, we further modified these sensors and packed them into a single transfection vector with high efficiency for the expression in both proliferative and differentiated cells, such as HEK293T, MEF, and rat adult cardiomyocytes. The majority of transfected cells (85%) co-express our dual-indicators, while 15% of cells are expressing one (11% mito-smacATPi only and 4% cyto-smacATPi only) (Appendix A). This is somewhat of a limitation, although the fluorescent trends seen via each indicator should remain representative of the biological actions occurring. More importantly, we excluded potential non-specific metabolic and growth effects on cells with high-level expression of the smacATPi. Additionally, ATP signals we observe in HEK293T cells are independent of off-target or background effects of smacATPi expression. Therefore, results from this study support the feasibility of the dual ATP indicator for the study of spatiotemporal dynamics of ATP in cytosol and mitochondria of cultured cells.

### 4.2. Real-Time Spatiotemporal Cellular ATP Dynamics in the Presence of Mitochondrial and Metabolic Inhibitors

Due to the constraints of currently available and affordable technology, the real-time effects of metabolic changes on cellular ATP dynamics remain ill-defined. The use of smacATPi may help solve some of these puzzles. We here demonstrated how cytosolic and mitochondrial ATP are altered in response to inhibitors of cytosolic glycolysis and mitochondrial ATP synthesis, at least in cultured HEK293T cells. Cytosolic ATP dramatically plummeted as expected in response to 2-DG under normoxic conditions. Interestingly, mitochondrial ATP also decreased by about 15%, suggesting that the inhibition of cytosolic glycolysis limits pyruvate production and its further oxidation in mitochondria. Additionally, our discoveries show that seemingly high concentrations of drugs are required to produce a response, which might be due to the instrumentation used or the sensitivity of smacATPi. 

Consequently, mitochondrial ATP becomes the primary but declined ATP source. As expected, mitochondrial ATP decreased significantly under oligomycin treatment. These results suggest that glycolytic activity in the cytosol may be increased to produce ATP that replenishes the mitochondrial pool via reversed transportation, as previously proposed [11,12]. These results indicate that ATP pools within these two cellular compartments are tightly coordinated, potentially via cross-compartment trafficking. 

### 4.3. ATP Regulation under Hypoxic Conditions

Oxygen is an essential element in OXPHOS [13,14,15,16]. Hypoxia triggers dramatic cellular reprogramming to gain ATP and maintain metabolic function and survival [17,18,19,20,21,22]. Hypoxia increases glycolytic ATP production via the enzymatic activity of phosphofructokinase-1 and pyruvate kinase. When oxygen concentrations are low for extended amounts of time, cells activate hypoxia-inducible factors (HIFs) [21,22,23,24], which bind to hypoxia-responsive elements (HREs), activating gene transcription of glycolysis [25,26]. However, exact changes in ATP dynamic and trafficking in real time between the cytosol and mitochondria in cells subjected to hypoxia remain vague [19,20,21,22,27]. Real-time evaluation of ATP dynamics in hypoxic conditions can help further define how the upregulation of cytosolic glycolysis maintains cell survival in hypoxia. 

Our finding is unexpected in that mitochondrial ATP increases and cytosolic ATP decreases in cells subjected to hypoxia. It appears that under hypoxic conditions, cells are highly adaptive and prioritize sustaining and increasing the mitochondrial ATP content, at least during the initiated two-hour window. This may occur via reverse transportation from the cytosolic ATP pool [28,29]. This observation appears to support the conclusions from early studies based on mitochondrial membrane potential changes suggesting that mitochondria become ATP consumers under hypoxic conditions [30,31]. Cytosolic ATP enters mitochondria, where complex V hydrolyzes ATP to reversely pump proton from the matrix to the intermembrane space, preventing the membrane potential collapse and the subsequent cell death. However, it remains a mystery how the cytosolic ATP is imported from the cytosol to the mitochondria. 

It is well-recognized that the ATP/ADP carrier (AAC) is critical in transferring ADP to mitochondria for ATP synthesis and ensuring that the cytosol receives adequate ATP from mitochondria to supply energy to the rest of the cell [32,33,34]. In the IMM, AAC is responsible for the uniport-style shuttling of ATP from the mitochondrial matrix to the intermembrane space and ADP from the intermembrane space to the mitochondrial matrix [35] (Figure 4A). When inhibited, the carrier protein can no longer transport ADP/ATP appropriately. Atractyloside (ATR), a toxic glycoside, locks AAC in the c-state, where the substrate-binding site is accessible to the intermembrane space [36,37,38] (Appendix A). This inhibition allows ADP to enter the carrier protein and bind but does not allow for its transportation into the matrix from the cytosol [6]. Our results show that ATR treatment in normoxic cells attenuates mitochondrial and, to a less extent, cytosolic ATP, suggesting that ATR-related inhibition contributes to mitochondrial ATP decline [39]. Our finding largely supports the previously proposed equimolar ADP–ATP exchange model [40,41]. Inhibition of AAC may increase cytosolic [ADP] and activate glycolysis, leading to slightly elevated cytosolic ATP. 

We unexpectedly found that cells in hypoxia showed elevated mitochondrial ATP and decreased cytosolic ATP. We suspect that hypoxia triggers the reversed ATP transportation into mitochondria, at least at the initial stage. The exact pathways mediating the reverse ATP transportation remain elusive. It has been suggested that AAC could reversely translocate ATP from the cytosol to mitochondria in hypoxia to sustain mitochondrial membrane potential via proton replenishing from the matrix, driven by ATP synthase-mediated ATP hydrolysis [42]. However, the treatment of ATR to inhibit AAC seems to exert little effect on mitochondrial ATP but slightly increased cytosolic ATP in hypoxia, suggesting that ATP may be derived more from the upregulated cytosolic glycolysis and somehow replenish the mitochondria via AAC-independent mechanisms. When 2-DG was added in addition to ATR in hypoxia, the cytosolic rescue seen with ATR alone was gone, leaving cytosolic values lower.

Interestingly, the addition of 2-DG decreased the hypoxic-induced mitochondrial ATP increase, thus bringing the mitochondrial ATP levels down to baseline. This is probably due to glycolysis inhibition—the primary ATP source in hypoxia. Without this additional ATP source, the mitochondrial ATP stores cannot be “rescued”. A recent study on mitochondrial assessment suggests that mitochondrial import of glycolytic ATP is independent of AAC, at least in cultured cancer cells [43,44]. However, results from the current investigation suggest that AAC at least partially mediated the reversed ATP transportation. Other AAC-independent pathways should also contribute to the reversed ATP transportation in hypoxia. Further studies are required to identify these pathways. The use of our novel dual ATP biosensor should help facilitate the exploration. 

## 5. Conclusions

In summary, our study demonstrates that smacATPi allows spatiotemporal recording of ATP dynamics simultaneously in mitochondria and cytosol of cultured cells, enabling real-time monitoring of cellular ATP dynamics in cytosol and mitochondria in response to metabolic changes under normoxia and hypoxia. This innovative technology will enable in-depth investigations to define cellular energetic regulations and gain further mechanistic insights into cellular energy metabolism in health and disease. 

## Figures and Tables

**Figure 1 cells-12-00695-f001:**
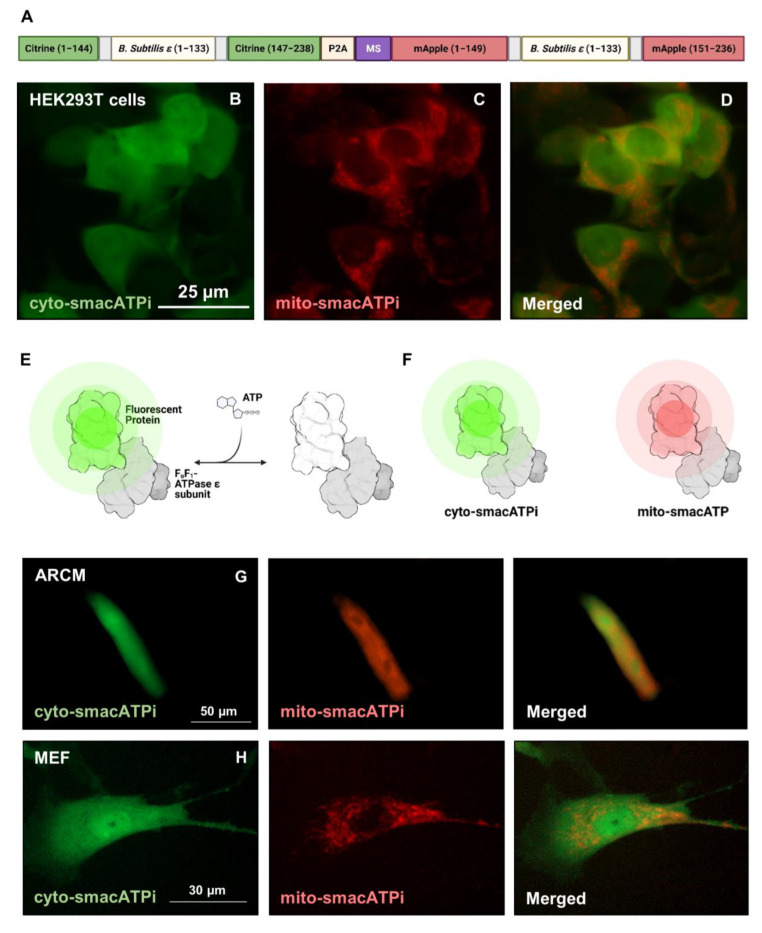
Dual-ATP indicator (smacATPi) design, testing, and confirmation of viability. (**A**) The simultaneous mitochondrial and cytosolic ATP indicator (smacATPi) was initially based on the previously described individual cytosolic (cyto-smacATPi) and mitochondrial (mito-smacATPi) ATP indicators, with a mitochondrial signaling sequence (MS) included for mitochondrial localization. smacATPi contains the polycistronic arrangement of sequences coding for these two MaLion fluorescent fusion proteins, along with a P2A region allowing for proper protein cleavage. (**B**) When smacATPi is expressed, the cytosolic indicator (cyto-smacATPi) indicates cytosolic ATP. (**C**) The mitochondrial indicator (mito-smacATPi) indicates mitochondrial-specific ATP. (**D**) When the indicators’ fluorescence is overlapped in living cells, ATP production and dynamic flux can be evaluated between both cellular compartments in vitro. (**E**) ATP binds to the epsilon subunit of F_1_F_o_-ATP synthase, triggering a conformational change in the connected protein, causing fluorescence. (**F**) The two indicators expressed from a single vector allow for the dual-compartmental visualization of ATP. (**G**,**H**) Representative images of ATP in adult rat cardiomyocytes (ARCM) and in mouse embryonic fibroblasts (MEF).

**Figure 2 cells-12-00695-f002:**
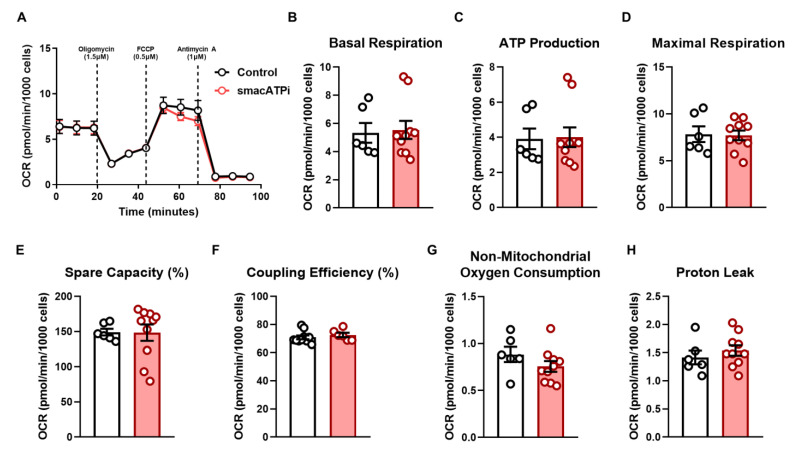
The expression of smacATPi does not interfere with cellular respiration. (**A**) Cellular bioenergetics was assessed in HEK293T cells with smacATPi expression. The oxygen consumption rate (OCR) of the cells transfected with smacATPi showed no change compared to non-transfected cells, indicating no adverse effects of smacATPi on cellular metabolism. (**A**) Overall OCR in response to inhibitors of oxidative phosphorylation. (**B**) Basal respiration (OCR) rate. (**C**) ATP-linked respiration (ATP production rate). (**D**) Maximal respiration. (**E**) Spare respiratory capacity. (**F**) Coupling efficiency. (**G**) Non-mitochondrial oxygen consumption. (**H**) Proton leak respiration. No change of the above parameters between the non-transfected HEK293T cells and those transfected with smacATPi could be detected (*n* > 6; error bars are SEM).

**Figure 3 cells-12-00695-f003:**
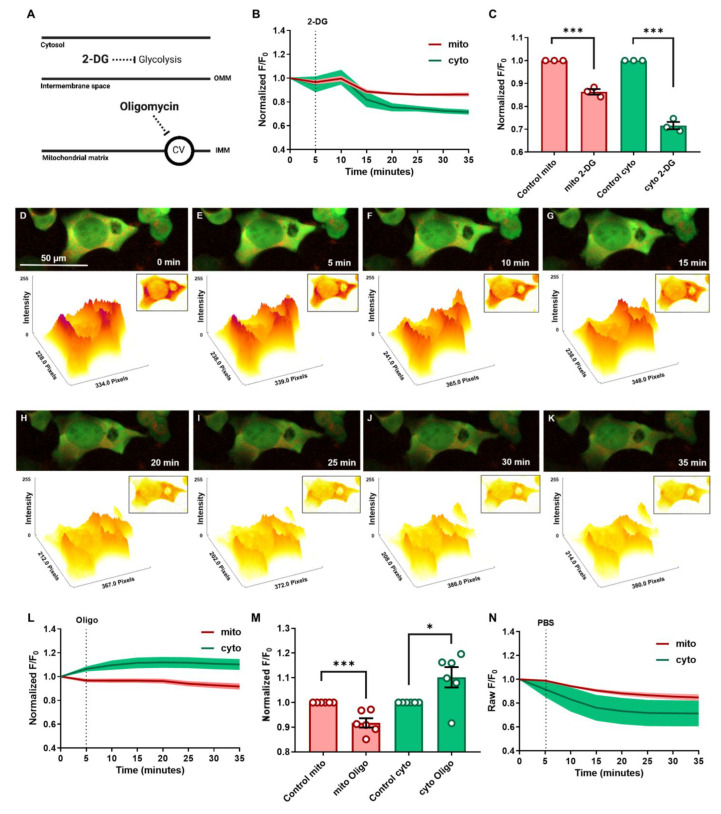
Spatiotemporal ATP dynamics in mitochondria and cytosol can be evaluated in real time using smacATPi after treatment with glycolytic and ATP synthase inhibitors. (**A**) Illustration of the inhibitory effect of 2-deoxyglucose (2-DG) on glycolysis in the cytosol and oligomycin on Complex V of the electron transport chain within the inner mitochondrial membrane. (**B**,**C**) The effect of 2-DG on the fluorescent intensity of ATP in mitochondria (red) and cytosol (green) on cultured HEK293T cells was graphed over time, pre- and post-drug administration. (**C**) Maximal changes of mitochondrial and cytosolic fluorescent intensity in response to 2-DG treatment. (**D**–**K**) The HEK293T cell images are from a respective 25 mm 2-DG drug treatment experimental run. The image below each fluorescent image is a FIRE LUT version of the respective image, allowing for the investigation of pixel intensity via surface plot. (**L**,**M**) Oligomycin (100 µM) was also used to inhibit mitochondrial ATP production in the HEK293T cells. (**N**) The fluorescent intensity was plotted over time and normalized to PBS treatment (*n* = 3 or more fields, with 30–90 cells analyzed in total; error bars are SEM; * *p* ≤ 0.05, *** *p* ≤ 0.001).

**Figure 4 cells-12-00695-f004:**
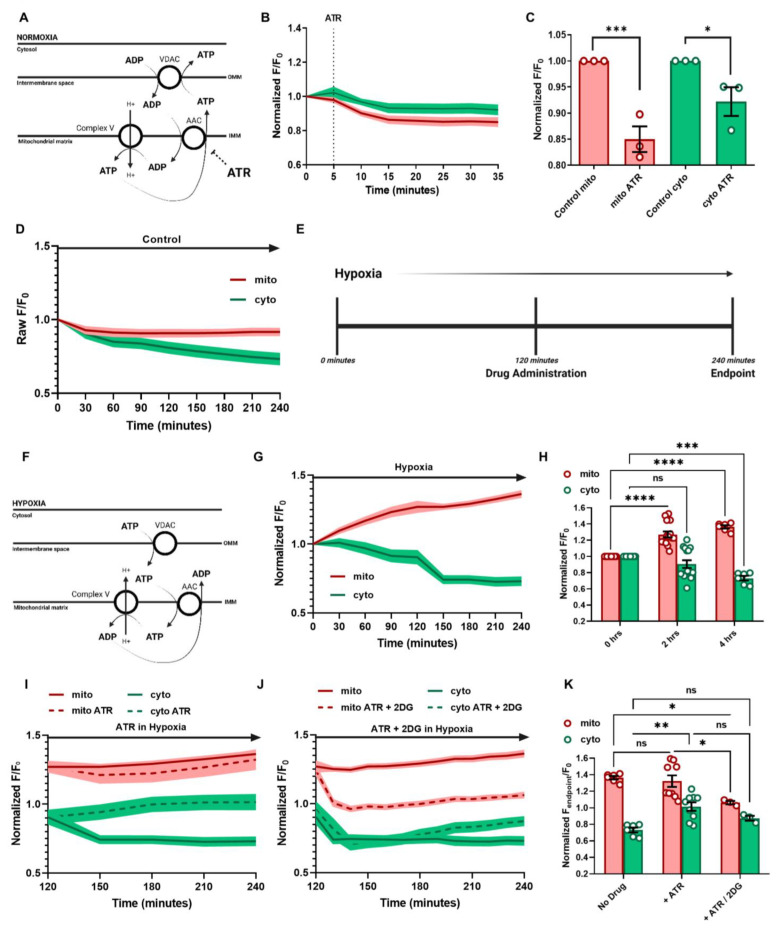
Effects of ADP/ATP Carrier (AAC) inhibition on spatiotemporal ATP dynamics in cultured HEK293T cells under normoxic and hypoxic conditions. (**A**) Under normoxic conditions, AAC facilitates the movement of ATP out of the matrix, while also bringing ADP to feed the ATP synthesis. (**B**,**C**) smacATPi-transfected HEK293T cells were treated with atractyloside (ATR), an AAC inhibitor, under normoxia. (**D**) A 4-h normoxic control run was used to normalize the hypoxia cell runs. (**E**) The protocol for hypoxia experiments involves 2 h of hypoxia exposure (2% O_2_, 5% CO_2_), drug administration, then an additional 2 h in hypoxia. (**F**) The previously proposed function of AAC in the reverse ATP transportation in hypoxia. (**G**,**H**) smacATPi-transfected cells were subjected to 4 h of hypoxia. The fluorescent intensity was normalized to the raw fluorescent intensity shown in (**D**). (**I**,**J**) Normalized fluorescent intensity in response to ATR and 2-DG + ATR treatments in cultured HEK293T cells during the hypoxic period. (**J**) The maximal effects of ATR + 2DG on mitochondrial and cytosolic ATP over the hypoxic period. (**K**) The endpoint fluorescent intensity between no drug, ATR, and ATP+2-DG drug runs. (*n* = 3 or more fields, with 30–90 cells analyzed in total; error bars are SEM; ns *p* > 0.05, * *p* ≤ 0.05, ** *p* ≤ 0.01, *** *p* ≤ 0.001, **** *p* ≤ 0.0001).

## Data Availability

Data can be made available via contacting corresponding author.

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
