# Peer review of "Real-Time Visualization of Cytosolic and Mitochondrial ATP Dynamics in Response to Metabolic Stress in Cultured Cells"

_cells, 2023, doi:10.3390/cells12050695_

Round 1

Reviewer 1 Report

The manuscript by White et al. presents an improvement of the previously described genetically-encoded ATP sensors developed by the laboratory of Kitaguchi. The originality of the work is the production of smacATPi, a dual-ATP indicator for both mitochondria and cytosol, the two indicators being encoded by a single plasmid and linked  by a short P2A self-cleaving peptide. Although potentially very interesting, the work suffers from a series of flaws and errors that lower its value and makes it barely convincing. It is the opinion of the reviewer that a better care at the design and realization of experiments can allow this work to achieve its potential.

One main issue with this work is that no experiment is showing the efficiency of the P2A self-cleavage, an absolute requirement to prove that the construct will not exist as a non cleaved form, or only in acceptable low proportions. The detection of a red faint signal in the cytosol of transfected cells can be considered as the presence of a non cleaved form (or likely the presence of out-of-focus light, a problem that can be resolved with the use of a confocal microscope). A western blot analysis is needed to address this question and also control correct localization of the cleaved forms. Alternatively, using confocal images, the authors could perform a colocalization assay and measure Pearson’s coefficient and Mandel’s coefficients and show exclusion of the two dyes. The authors should also address the question of the presence of cells with red signals but no green one. What is the proportion of the cells that display both detectable signals? The proportion of the cells displaying only one? Is the red signal completely colocalizing with another mitochondrial marker? The lack of analysis for those elementary questions of quality control doesn’t help convince readers of the quality of the work. This is the main difference compared to the previous paper on those sensors, and it should be much more detailed.

A similar lack of attention to the work can be felt when the introduction is simply duplicated (line 72 to 104) and no proof-reading has fixed this simple mistake.

Although the method is not quantitative per se, it could still be presented as semi-quantitative.

The authors should describe in more details the small mutations (point mutations?) induced (introduced?) for neg-smacATPi. Are those mutations interfering with expression, stability, cleavage and proper localization?

Line 288: “We confirmed that the mutated form of the indicator showed no fluorescence (Figure S2C)”. This is Figure S1C.

Legend for figure 1C is missing.

Are the fluorescence runs measured per field or per cell? The images and movies indicate that the variability of fluorescence between cells is too large to measure per field, however, this is not clarified in the text.

In figure S2, the vehicle is PBS, but is it really the correct vehicle? Deoxyglucose was diluted in PBS? Oligomycin too? ATR?

Figure S2 is puzzling for a more serious reason. Why is the addition of PBS inducing a decrease in fluorescence higher than the one measured without addition (S2B vs S2C)? When nothing is added, mitochondrial fluorescence stabilizes after 30 minutes around 90% of the original value but cytosolic fluorescence keeps decreasing: why? This alone suggests that there are experimental problems that need to be addressed and it makes the rest of the work unconvincing. Tacking pictures every 5 minutes should not result in a photobleaching that high.

The addition of PBS is inducing a decrease in fluorescence : Is Figure 3B normalized to this effect? It is very concerning that it does not seem to be the case and that adding 2deoxiglucose or PBS result in the SAME reduction. The vehicle effect should not be in a supplementary figure but directly in figure 3. Other drugs have been reported to better inhibits glycolysis, such as arsenic compounds or 3-bromopyruvate, and should be consider for testing.

The use of highlighted graphs is unclear: are those the limits of standard errors or simply the upper and lower traces of replicates?

In figure 3D, why using oligomycin alone and not have used oligomycin and atractyloside to isolate metabolically the mitochondria from the cytosol? The observations of figure 3D are not really explained. Maybe the use of HEK293 cells is not a good model. Those cancer cells are known to be glycolytic and not rely massively on OXPHOS. To fix that, cells could be acclimated to use galactose instead of glucose, to force them to rely on OXPHOS. Other cells types could be used that rely more on OXPHOS.

In figure 4C, the trace lines indicated as a legend concern figure 4G, so they should be placed in the right graph to avoid confusion. Experiments implicating atractyloside could also be done in presence of oligomycin and FCCP to see if the fluorescence can decrease lower than by just 15%. Again, the vehicle is already inducing a decrease that has a similar extent and it must be clear in the graph if the traces are normalized to that effect.

Figure S3: “Furthermore, upon ATR treatment on cultured HEK293T cells in hypoxia the mitochondrial potential slightly decreased and was also decreased in hypoxia conditions” Authors should comment more on those observations. If TMRM in hypoxia was sharply decreasing over the course of 24h , why not also follow fluorescence of smacATPi during the same period? What is the effect of atractyloside over longer periods than 35 minutes on smacATPi fluorescence, on TMRM fluorescence? It seems that atractyloside is not really working well at inhibiting AAC and authors should comment on those observations.

Figure S4D and S4F, at which time those values were calculated? Why the mito-pH signal keeps increasing over 240 minutes in normoxic conditions? Moreover, clearly the signal is changing a lot during the first 10 minutes, both in normoxia or hypoxia and this change must be explained and taking into account when calculating the F/F0 values. Fig S4F is misleading, the signals are very clear, the mito-pH signal keeps increasing over 240 minutes in normoxic conditions, but it is more or less stable in hypoxic conditions. Normalization at 240 minutes cannot make this observation disappear or only artificially because the apparent variability of the mito-pH signal in hypoxia is much higher than in normoxia. If anything, those measures are suggesting pH issues rather than rejecting them.

Reviewer 2 Report

Comments to the authors:

In the submitted manuscript the authors report the design and validation of a genetically encoded fluorescent ATP indicator that allows for real-time, simultaneous visualization of cytosolic and mitochondrial ATP content in cultured cells. Despite a strong effort in the characterization of the dual-ATP indicator, demonstration of its feasibility is not convincing.

The scientific methods applied was not properly conducted in some cases, especially in the case of mitochondrial membrane potential measurements. The description of the methodology is superficial and some conclusions drawn by the authors are not supported by statistical analysis performed.

Major Concerns:

1.      Regarding dual-ATP indicator control experiments: How do the authors explain that cytosolic ATP content drops by 30 % within 30 min in PBS and more than 20 % after 4-hour incubation in normoxic condition? (see Fig. S2). Please indicate raw fluorescence data on y axis instead F/F0 values.

2.      Regarding mitochondrial membrane potential measurements in hypoxia (Fig. S3): In intact cells TMRM fluorescence readouts are distorted by other factors than ΔψM, for example plasma membrane potential (PMP). Therefore decrease of TMRM signal reflects changes in PMP as well. Plus calibration of ΔψM is missing, please demonstrate experimental results depicting changes of TMRM fluorescence referred to completely depolarized mitochondria in hypoxia and after ATR treatment as well. Please include number of fields of view and total cell number examined.

3.      The authors claim that’ results shown in Fig.1 validate the feasibility and utility of dual-ATP indicator in both proliferative and post-differentiated cells.’ However, this claim is incorrect, since it is not supported by statistical analysis performed on images taken from large number of fields.  Pictures shown in panel D and E are only representative images from one field of view.

4.      Regarding Fig.3: Total number of cells and fields of view examined in measurements is not indicated. Please include it in the figure legend.

5.      The concentration of glycolytic (25 mM 2-DG) or mitochondrial inhibitors (100 microM oligomycin or ATR) applied is extremely high. What was the reason for it? Please indicate concentration of mitochondrial inhibitors applied (oligomycin, FCCP, antimycin A) in the assessment of cellular bioenergetics (see Fig. 2/A).

6.      How do the authors explain that there is a continuous increase in mitochondrial ATP content during 4 hour of hypoxia? The authors propose that the mitochondrial ATP content is increasing, at least during the four-hour window, potentially via reverse transportation from the cytosolic ATP reserve. However, the cytosolic origin of this increasing mitochondrial ATP content is not examined in hypoxic condition when glycolysis is inhibited by 2-DG. Please add further experimental data to the manuscript demonstrating the effect of glycolytic inhibition on increasing mitochondrial ATP content during 4 hour of hypoxia.

7.      Please demonstrate the effect of bongrekic acid on mitochondrial ATP content under hypoxic condition.

Thus, the publication of the manuscript in its present form is not recommended.

Reviewer 3 Report

Thanks for the very nice piece of work on ATP indicators.

When reading the manuscript, I found that the results are not always clearly explained.

Some of the issues are the following:

Introduction is duplicated.

The wavelengths indicated for the TMRM don’t even make sense for a fluorescent probe. Please correct.

Figure 1 panel C is not mentioned in the legend nor in the manuscript.

Line 289: Would it be Figure S1C?

Legend of Figure S2 does not provide sufficient detail to read the figure.

Line 435: Would it be Figure 3B-C?

Round 2

Reviewer 1 Report

The manuscript is better written and structured and the reviewer is pleased that their various comments have been taken into consideration. The only minor points remaining are the requirement for all raw data to be displayed, and a discussion about the limitations of the use of the single plasmid, considering that up to 11% of the mitochondrial signal comes from cells devoid of cytosolic signal. Commenting on such limitation will not lower the interest for this work, but simply put it in perspective. Details are described below:

Comment 1 has been partially answered, as the presence of red and green signals in cells simply indicate that the two genetically-encoded dyes are fluorescent, not that they have been cleaved. The red signal is not formally demonstrated to be solely in the mitochondrial compartment, although it appears so, but the authors should use another mitochondrial signal to demonstrate mitochondrial localization in their future experiments.

Comment 2. The authors replied that ‘In rare situations, only one signal could be seen” but if 85% of the transfected cells show both signals, 15% show only one (96-85= 11% with red only and 4% of green only). Is it that rare? It seems that there is a good proportion of red only cells, and the authors have to provide comment on this limitation.

Comment 9. Oligomycin must be solubilized in ethanol or DMSO, PBS is not the right choice and authors will have to take this into consideration in future experiments.

Comment 11: (Figure 3) The normalization issue seems to have been solved but the reviewer requires to see ALL the raw data at least in supplemental figures to have a better understanding of these raw data. For instance, present 2DG raw data in addition to the vehicle raw data (presented in panel N).

If the authors present that mitochondrial red signal is decreased in 2DG treated cells (figure 3C) then they need to comment, even briefly, on the origin of this decrease in the result section .

Same comments for the oligomycin treatment, the 10% decrease is statistically significative but this can hardly be called significantly decreased. The reviewer would have preferred that conclusions would be drawn directly in the results sections rather than to wait the conclusion section to read a comment on the increase of glycolysis as a compensatory effect to produce cytosolic ATP that can then enter the mitochondria and keep mitochondrial ATP levels suboptimal. Those observations were the reason it was suggested to treat cells with combos of drugs, such as atractyloside and oligomycin together to reveal the full effect of oligomycin on mitochondria ATP.

Figure 4. Comparison of panels I and J, strongly suggests that the maintenance of mitochondrial ATP levels was due to glycolysis. Same comment than before, drawing a conclusion in the result paragraph would have been appreciated, rather than to wait to reach the conclusion part to read that the experiments suggest that ATP transportation is not mediated by the carrier.

Reviewer 2 Report

The authors have not satisfactorily responded all problems/concerns raised by the reviewer.

Namely, concern 1 was not elucidated: raw fluorescence intensity values (F) are not equal to raw F/F0 values. Therefore Fig. 3N and 4D do not reflect raw/relative fluorescence intensities measured by the camera. Plus it is still unsolved why the cytosolic ATP content drops by 30 % within 30 min incubation in PBS.

Regarding Concern 5: Necessity of application of inhibitors (such as 2-DG and/or oligomycin) in extreme high concentration can be limiting for the feasibility and utility of the dual-ATP indicator smacATPi in intact cells.

 Despite of the concerns listed above the manuscript can be accepted for publication.

Reviewer 3 Report

The manuscript was improved accordingly.

Round 3

Reviewer 1 Report

The authors took care of the minor issues remaining and the manuscript is acceptable in his present form.